# Psychometric validation and cultural adaptation of executive functioning scale in Malaysian University students

Hao Yin[ID], Muhammad Syawal Amran[☉]*

Faculty of Education, Universiti Kebangsaan Malaysia, Bangi, Selangor, Malaysia

☉ This author contributed to the work equllly and should be regarded as co-first authors.
* syawal@ukm.edu.my

## Abstract

Addressing the lack of culturally-appropriate assessment tools, this study adapted and validated the Executive Functioning Scale (EFS) for use within a Malaysian university student population. Utilizing a sample of 629 university students, we conducted a comprehensive psychometric evaluation of the adapted scale. Exploratory factor analysis (EFA) supported a refined 50-item, six-factor structure that accounted for 65.87% of the total variance. Subsequent confirmatory factor analysis (CFA) revealed that a second-order hierarchical model—in which the six first-order factors loaded onto a single, higher-order Executive Functioning construct—provided an excellent fit to the data (CMIN/df = 1.447, CFI = .976, RMSEA = .026). Critically, its fit was virtually indistinguishable from a more complex first-order model, strongly favoring the hierarchical conceptualization on the basis of parsimony. The scale also demonstrated strong discriminative validity, effectively differentiating between high- and low-performing groups across all dimensions. Collectively, the findings establish the Malaysian-adapted EFS as a robust and valid instrument for assessing the multifaceted nature of executive functioning. This research provides the first psychometrically sound tool for measuring executive functioning tailored to this population, offering a critical resource for future cross-cultural research, clinical diagnostics, and educational interventions. The validated scale facilitates a more nuanced understanding of cognitive constructs within a non-Western higher education context.

## Introduction

Executive functioning (EF) represents a suite of higher-order cognitive processes essential for goal-directed behavior and environmental adaptation [1]. These functions—encompassing core components like working memory, cognitive flexibility (set-shifting), and inhibitory control [2] —operate as the brain's "air traffic control system," orchestrating thought and action to navigate complex tasks and social landscapes [3]. The pivotal role of robust EF is extensively documented, serving as

**Data availability statement:** All relevant data are within the paper and its Supporting information files.

**Funding:** The author(s) received no specific funding for this work.

**Competing interests:** The authors have declared that no competing interests exist.

a powerful predictor of critical life outcomes, including academic attainment, occupational success, and overall mental and physical well-being [4]. Given its centrality to human cognition, the precise and valid measurement of EF has become a cornerstone of inquiry across clinical psychology, neuropsychology, and educational science [5].

Despite its importance, the assessment of EF is fraught with methodological challenges, most notably the issue of cross-cultural validity [6]. The vast majority of established EF assessment tools, including both performance-based tasks and self-report inventories, were conceived and standardized within Western, Educated, Industrialized, Rich, and Democratic (WEIRD) populations [7]. This origin introduces a critical problem: these instruments are inherently imbued with cultural norms about communication, problem-solving, and self-regulation that may not align with non-Western cultural values, particularly those prevalent in Asia [8]. For example, Western-centric scales often implicitly reward individualism and direct self-expression, which can conflict with cultural scripts emphasizing collectivism, hierarchical respect, and indirect communication to maintain social harmony [7,8].

This cultural dissonance can lead to a significant misinterpretation of abilities, potentially generating "false deficits." When a student's behavior is guided by culturally normative values, such as deference to authority or cautious decision-making, it risks being misconstrued as a lack of initiative or poor processing speed by a tool not attuned to these nuances. Such biased assessments carry profound consequences [9]. Executive functioning is intrinsically linked to academic success, stress management, and adaptability; therefore, interventions based on flawed data may be ineffective or even detrimental because they do not reflect students' genuine challenges [4]. Furthermore, as universities increasingly pursue global standards of equity and inclusivity, the use of culturally uncalibrated tools undermines fairness and can perpetuate stereotypes about cognitive performance across cultures. Achieving robust transcultural validity thus requires more than simple translation; it demands a rigorous process of adaptation and validation to ensure that an instrument is both conceptually and contextually appropriate [10].

This psychometric void is particularly pronounced in culturally diverse, non-Western settings such as Malaysia [11]. The Malaysian higher education landscape imposes substantial cognitive and self-regulatory demands on students, who must manage intricate academic workloads, engage in sophisticated critical thinking, and navigate a uniquely multicultural social milieu [12]. A nuanced understanding of their executive functioning is therefore indispensable for designing effective academic support systems and mental health services. Despite this clear imperative, there remains a significant scarcity of psychometrically robust instruments for assessing EF that have been specifically validated for the Malaysian university student demographic [13]. Researchers and clinicians in the region are consequently faced with a difficult choice: employ original English-language tools of unverified validity or rely on informal translations lacking systematic empirical scrutiny.

The Executive Functioning Scale (EFS) is a comprehensive inventory designed to capture multiple key facets of EF, with a theoretical structure comprising dimensions

such as Working Memory, Response Inhibition, and Set Shifting [14]. While this structure is well-supported in its original context [14], its psychometric integrity and theoretical coherence have yet to be examined within the Malaysian population. It remains an open empirical question whether the factor structure identified in Western samples will replicate, or whether a hierarchical model—which posits that these distinct EF components reflect a single, overarching latent construct—provides a valid representation of executive functioning in Malaysian students. This absence of foundational psychometric research constitutes a significant barrier to advancing the scientific study of cognition in Malaysia and curtails the development of evidence-based educational and clinical practices.

## The present study

To address these critical gaps, the present study undertook a comprehensive cultural adaptation and psychometric validation of the Executive Functioning Scale for use with Malaysian university students. Our investigation was designed to first establish the scale's fundamental properties, including item-level performance and internal consistency. The primary objective, however, was to elucidate the underlying factor structure of the adapted scale using a sequential approach of exploratory and confirmatory factor analyses. Critically, we aimed to test the tenability of a hierarchical, second-order factor model against plausible alternative models to identify the most theoretically sound and empirically supported conceptualization of EF in this unique population. In doing so, this study provides the first rigorously validated instrument for measuring executive functioning in Malaysian university students, establishing an essential foundation for future research and evidence-based applications in the region.

## Methodology

### Research design

The research involved two rounds of testing with different participant sets. In the first round, the Executive Functioning Scale (EFS) was administered to adolescents selected through random sampling to test its reliability and validity. In the second round, a subset of participants from the first round was randomly chosen for a retest two weeks later. This retest aimed to assess the test-retest reliability of the EFS, minimizing the impact of external events and practice effects.

### Research participants

To culturally adapt and psychometrically validate the Executive Functioning Scale (EFS), we conducted an online survey with Malaysian university students. Our initial distribution of 700 questionnaires yielded 629 valid responses for the primary analysis (Sample 1), representing an 89.85% effective response rate. As detailed in Table 1, this cohort was well-balanced by gender (312 males, 49.6%; 317 females, 50.4%) and academically diverse, comprising 296 students (46.8%) from Humanities and Social Sciences (e.g., Philosophy, Economics, Law) and 333 (53.2%) from STEM disciplines (e.g., Science, Engineering, Agriculture, Medicine). The sample also encompassed 308 undergraduate and 321

**Table 1. Basic data of the sample 1.**

| Variable Name | Types | Number |
|---|---|---|
| Gender | Male | 312 |
| | Female | 317 |
| Major | Humanities and Social Sciences | 296 |
| | STEM | 333 |
| Educational background | Undergraduate | 308 |
| | Postgraduate | 321 |

postgraduate students. To assess test-retest reliability, we recruited a subsample of 300 participants (Sample 2) from this cohort for a follow-up survey two weeks later. This second wave of data collection achieved a 100% response rate, and the resulting sample was also gender-balanced, consisting of 151 males (50.3%) and 149 females (49.7%). Collectively, these samples form a robust and representative foundation for the ensuing psychometric evaluation.

### Ethical oversight and informed consent protocol

This cross-sectional investigation, spanning May 1 to August 28, 2024, engaged undergraduate students at Malaysian public universities via a structured online survey. Formal ethics clearance (Ethic Approval No: KPM.600–3/2/3-eras(20340)) was secured on May 1, 2024, from the Bioethical Committee at Universiti Kebangsaan Malaysia's Faculty of Education, ensuring adherence to Declaration of Helsinki principles. Electronic written informed consent was systematically implemented: prospective participants first encountered a digitally embedded disclosure form detailing research objectives, voluntary participation terms, data anonymization protocols, and unrestricted withdrawal rights. Explicit confirmation via mandatory checkbox ("I consent to participate") served as the gateway to survey access, with all responses irrevocably anonymized. The ethics review body formally validated this electronic consent mechanism, eliminating the necessity for handwritten signatures. Notably, the study cohort exclusively comprised adults, precluding any requirement for guardian consent.

### Research instruments

Executive Functioning Scale (EFS) This study utilized the Executive Functioning Scale (EFS), developed by Thomas W., Mirko Uljarević, et al and adapted for self-report [14]. The EFS, designed to evaluate key aspects of executive functioning in children and adolescents, comprises 52 items based on self-reported experiences.

The scale employs a five-point Likert scale ranging from 'never' to 'very often' and covers six major components of executive functioning: Working Memory (items 1–7, 42–46), Risk Avoidance (items 8–12, 21, 38), Response Inhibition (items 29–33, 39–41), Emotion Regulation (items 14–20, 22–24), Goal Shifting (items 34, 36–37, 47–52), and Processing Speed (items 25–28). It also assesses the overall level of executive functioning (items 1–52). Higher total scores in each dimension indicate better executive functioning.

To ensure the psychometric integrity and ecological validity of the Executive Functioning Scale (EFS) for Malaysian university students, a rigorous, multi-stage cultural adaptation was undertaken. The primary objective was to transcend literal translation and achieve deep conceptual and contextual equivalence, thereby addressing the potential cultural biases inherent in Western-developed instruments. This process systematically incorporated best practices for cross-cultural research.

First, to establish linguistic equivalence, a standardized forward-backward translation protocol was employed. The original 52-item English EFS was translated into Malay by a bilingual expert. A second, independent bilingual expert, blind to the original instrument, then back-translated the Malay version into English. The original and back-translated English versions were meticulously compared to identify and reconcile any semantic or idiomatic discrepancies, ensuring the linguistic fidelity of each item.

Second, to ensure conceptual and contextual equivalence, the adapted items were critically reviewed for their relevance to the lived experiences of Malaysian university students. This involved adapting items referencing culturally specific scenarios. For instance, an original item assessing complex instruction-following, "Skilled at following complex instructions with more than 8 steps (e.g., baking, constructing objects)," was revised to enhance its ecological validity. It was rephrased to reflect a salient academic context: "I am effective at following a multi-stage research protocol or laboratory procedure." This modification anchors the assessment in a higher-order academic task directly aligned with the cognitive demands placed upon this population.

Furthermore, developmental appropriateness was a key consideration. Items originally designed for younger adolescents were re-engineered to capture the cognitive sophistication required in higher education. For example, an item assessing simple planning was reframed to probe a student's ability to "Effectively manage and prioritize tasks across multiple courses with competing deadlines," thus ensuring the scale possessed sufficient ceiling to differentiate among high-achieving young adults.

Finally, to incorporate local norms and expert input, an expert panel comprising two local educational psychologists and one linguist reviewed the entire adapted scale. This panel provided critical feedback on the cultural appropriateness, clarity, and conceptual relevance of each item. For instance, an item related to emotional regulation, "Seems overly sensitive to criticism," which is susceptible to cultural interpretation, was rephrased to be more behaviorally anchored and context-specific: "I find it difficult to remain objective when receiving critical feedback on my academic work." This comprehensive process, integrating expert review with contextual and developmental adaptation, established a robust foundation for the scale's subsequent psychometric validation within the Malaysian university context.

## Results

### Item analysis

To evaluate the internal consistency and item relevance of the scale, we conducted an item-total correlation analysis using data from Sample 1. Each item's score was correlated with its corresponding subscale score and the total scale score. All item-subscale and item-total correlations were statistically significant ($p < .01$). Notably, while items 13 ("I can make good decisions in dangerous situations") and 35 ("I can come up with new activities or things to do") did not load uniquely onto a single dimension, their conceptual relevance to overall executive functioning supported their inclusion. Consequently, 50 original items were retained for the adapted scale.

We further assessed the scale's discriminant validity via an extreme group comparison. Participants from Sample 1 were ranked by their total scores, with the top 27% and bottom 27% designated as the high- and low-scoring groups, respectively. Independent samples t-tests were then performed to compare the mean scores between these groups. As detailed in Table 2, the high-scoring group (M = 242.78, SD = 7.30) demonstrated a significantly higher mean total

**Table 2. Tests of item differentiation.**

| Variable name | Group | M±SD | F | Sig | t | Sig | Mean Difference | 95% CI (From Lower to Upper) |
|---|---|---|---|---|---|---|---|---|
| Working Memory | Low Group | 2.42±0.45 | 264.93 | <0.05 | −71.83 | < 0.05 | −2.42 | −2.49~−2.36 |
| | High Group | 4.84±0.11 | | | | | | |
| Risk Avoidance | Low Group | 2.21±0.46 | 242.51 | < 0.05 | −75.16 | < 0.05 | −2.65 | −2.72~−2.58 |
| | High Group | 4.86±0.12 | | | | | | |
| Response Inhibition | Low Group | 2.52±0.44 | 210.39 | < 0.05 | −71.29 | < 0.05 | −2.37 | −2.44~−2.31 |
| | High Group | 4.89±0.10 | | | | | | |
| Emotional Regulation | Low Group | 2.30±0.39 | 188.31 | < 0.05 | −85.09 | < 0.05 | −2.54 | −2.60~--2.48 |
| | High Group | 4.84±0.12 | | | | | | |
| Set Shifting | Low Group | 2.49±0.47 | 283.93 | < 0.05 | −67.14 | < 0.05 | −2.38 | −2.46~--2.31 |
| | High Group | 4.87±0.11 | | | | | | |
| Processing Speed | Low Group | 2.20±0.52 | 221.13 | < 0.05 | −67.18 | < 0.05 | −2.67 | −2.74~−2.59 |
| | High Group | 4.87±0.14 | | | | | | |
| Executive Functioning | Low Group | 140.32±18.36 | 150.01 | < 0.05 | −70.56 | < 0.05 | −102.46 | −105.33~--99.60 |
| | High Group | 242.78±7.30 | | | | | | |

score than the low-scoring group (M = 140.32, SD = 18.36; t = −70.56, p < .001). This significant difference was consistently observed across all six subscales—Working Memory, Risk Avoidance, Response Inhibition, Emotional Regulation, Set Shifting, and Processing Speed—with the high-scoring group outperforming the low-scoring group in all instances (p < .001). These results provide robust evidence for the scale's ability to effectively differentiate between individuals with varying levels of executive functioning, thereby supporting its discriminant validity and justifying subsequent factor analysis.

## Reliability

**Internal consistency reliability.** For the Executive Functioning Scale (EFS), the Cronbach's alpha coefficient for the total questionnaire score was 0.96. The alpha coefficients for the six factors—Working Memory, Risk Avoidance, Response Inhibition, Emotion Regulation, Set Shifting, and Processing Speed—were 0.94, 0.91, 0.91, 0.93, 0.92, and 0.84, respectively.

*Test-retest reliability.* Based on the retest data for Sample 2, the test-retest reliability of the total questionnaire score, assessed two weeks later, was over 0.90. The test-retest reliability coefficients for the six factors—Working Memory, Risk Avoidance, Response Inhibition, Emotion Regulation, Set Shifting, and Processing Speed—were 0.88, 0.92, 0.87, 0.89, 0.84, and 0.86, respectively.

## Construct validity

**Exploratory factor analysis.** After excluding the two items that did not belong to any dimensions, a feasibility test for factor analysis was conducted on the 50 items of the Executive Functioning Scale (EFS) in Sample 1. The results indicated that the Kaiser-Meyer-Olkin (KMO) measure of sampling adequacy for the EFS questionnaire was 0.971, and Bartlett's test of sphericity was statistically significant ($\chi^2$ = 20355.024, df = 1225, (P < 0.01)), suggesting that the data were suitable for factor analysis. Specifically, the KMO value is used to measure the suitability of the data for factor analysis and is generally interpreted as follows: 0.6–0.7: acceptable; 0.7–0.8: moderate; 0.8–0.9: good; above 0.9: excellent. Bartlett's Test of Sphericity is used to test whether there is sufficient correlation between variables to proceed with factor analysis. The significance value (p-value) should be less than 0.05, indicating significant correlations, allowing factor analysis to continue. The primary goal of factor analysis is to extract enough factors to explain most of the variance in the data. It is generally considered ideal if the cumulative variance explained exceeds 60%. Factor loadings reflect the contribution of each variable to the corresponding factor. It is generally considered that a loading greater than 0.4 or 0.5 has practical significance [15,16].

To establish the construct validity of the adapted scale, we performed an exploratory factor analysis (EFA) on Sample 1 data using principal component analysis with a varimax orthogonal rotation. An iterative process was employed, removing items with factor loadings below.50 or those belonging to factors with fewer than three items. This procedure resulted in a final 50-item scale with a clear six-factor solution, where all factors had eigenvalues greater than 1 and collectively accounted for 62.99% of the total variance. As shown in Table 3, all retained items exhibited strong primary factor loadings, ranging from.63 to.86. These factors were identified as Working Memory, Risk Avoidance, Set Shifting, Response Inhibition, a second distinct Risk Avoidance factor, and Processing Speed. Collectively, these results provide robust evidence for the construct validity of the 50-item adapted scale and confirm its underlying six-factor structure within this population.

**Confirmatory factor analysis.** To rigorously evaluate the construct validity of the Malaysian Executive Functioning Scale (EFS), we conducted a Confirmatory Factor Analysis (CFA) on Sample 1 (N = 629) using AMOS software. This analysis was predicated on the six-factor structure identified in a prior exploratory factor analysis (EFA), with the primary objective of determining the optimal latent structure by comparing a series of theoretically grounded, competing models.

**Table 3. EFS exploratory factor analysis results (standardised regression coefficients).**

| Working Memory | | Emotional Regulation | | Set Shifting | | Response Inhibition | | Risk Avoidance | | Processing Speed | |
|---|---|---|---|---|---|---|---|---|---|---|---|
| Item | Loadings | Item | Loadings | Item | Loadings | Item | Loadings | Item | Loadings | Item | Loadings |
| Item 1 | 0.863 | Item 14 | 0.759 | Item 34 | 0.759 | Item 29 | 0.818 | Item 8 | 0.843 | Item 25 | 0.660 |
| Item 2 | 0.736 | Item 15 | 0.727 | Item 36 | 0.723 | Item 30 | 0.754 | Item 9 | 0.790 | Item 26 | 0.718 |
| Item 3 | 0.770 | Item 16 | 0.769 | Item 37 | 0.747 | Item 31 | 0.754 | Item 10 | 0.651 | Item 27 | 0.705 |
| Item 4 | 0.723 | Item 17 | 0.765 | Item 47 | 0.755 | Item 32 | 0.755 | Item 11 | 0.723 | Item 28 | 0.791 |
| Item 5 | 0.663 | Item 18 | 0.667 | Item 48 | 0.636 | Item 33 | 0.747 | Item 12 | 0.782 | – | – |
| Item 6 | 0.735 | Item 19 | 0.806 | Item 49 | 0.660 | Item 39 | 0.671 | Item 21 | 0.788 | – | – |
| Item 7 | 0.762 | Item 20 | 0.729 | Item 50 | 0.653 | Item 40 | 0.701 | Item 38 | 0.751 | – | – |
| Item 42 | 0.771 | Item 22 | 0.840 | Item 51 | 0.824 | Item 41 | 0.730 | – | – | – | – |
| Item 43 | 0.753 | Item 23 | 0.824 | Item 52 | 0.809 | – | – | – | – | – | – |
| Item 44 | 0.762 | Item 24 | 0.659 | – | – | – | – | – | – | – | – |
| Item 45 | 0.753 | – | – | – | – | – | – | – | – | – | – |
| Item 46 | 0.750 | – | – | – | – | – | – | – | – | – | – |

We specified and systematically evaluated three distinct models to this end. The first was a first-order model (FOV) that posited six correlated latent factors representing the dimensions of executive functioning, each defined by its respective set of observed items (see Fig 1). The second, a more theoretically elegant second-order model (SOV), proposed a hierarchical framework where these six first-order factors are themselves indicators of a single, higher-order 'Executive Functioning' construct (see Fig 2). This SOV model offers a more parsimonious account of the substantial inter-correlations observed among the six primary dimensions. The third model (FOV + CMV) was specifically designed to assess the potential influence of Common Method Variance (CMV), a prevalent concern in self-report data. This model augmented the FOV by introducing a general method factor, to which all items loaded, allowing us to isolate and quantify potential systematic method bias (see Fig 3).

Model adequacy was assessed against a comprehensive suite of stringent fit indices: the chi-square to degrees of freedom ratio (CMIN/df < 3), Comparative Fit Index (CFI > 0.9), Tucker-Lewis Index (TLI > 0.9), Goodness of Fit Index (GFI > 0.9), Adjusted Goodness of Fit Index (AGFI > 0.9), Root Mean Square Error of Approximation (RMSEA < 0.05), and Standardized Root Mean Square Residual (SRMR < 0.05).

The analysis, detailed in Table 4, revealed that all three competing models demonstrated an excellent fit to the data, with key indices comfortably exceeding established psychometric benchmarks. Specifically, the fit indices for the first-order model (CMIN/df = 1.431, CFI = .975, RMSEA = .026) were virtually indistinguishable from those of the second-order model (CMIN/df = 1.430, CFI = .974, RMSEA = .026). This equivalence in fit is a pivotal finding, as it provides compelling support for the more parsimonious, theoretically sophisticated SOV model, showing it can explain the shared variance among the factors as effectively as the less constrained FOV model. A detailed examination of the parameter estimates further solidified this conclusion. Within the SOV, all six first-order factors exhibited strong and statistically significant standardized loadings onto the higher-order 'Executive Functioning' factor, ranging from .73 to .79, substantiating its role as an overarching construct. Furthermore, the individual items displayed robust loadings on their designated first-order factors (ranging from .69 to .84), confirming their validity as indicators of their respective sub-dimensions. Finally, our evaluation of method bias showed that while the FOV + CMV model produced a marginal improvement in fit (CFI = .980), the change was negligible, thereby ruling out CMV as a significant confounding variable.

In aggregate, guided by the principle of parsimony and the compelling empirical evidence for a hierarchical structure, the second-order model was retained as the most veridical and theoretically sound representation of the EFS in this

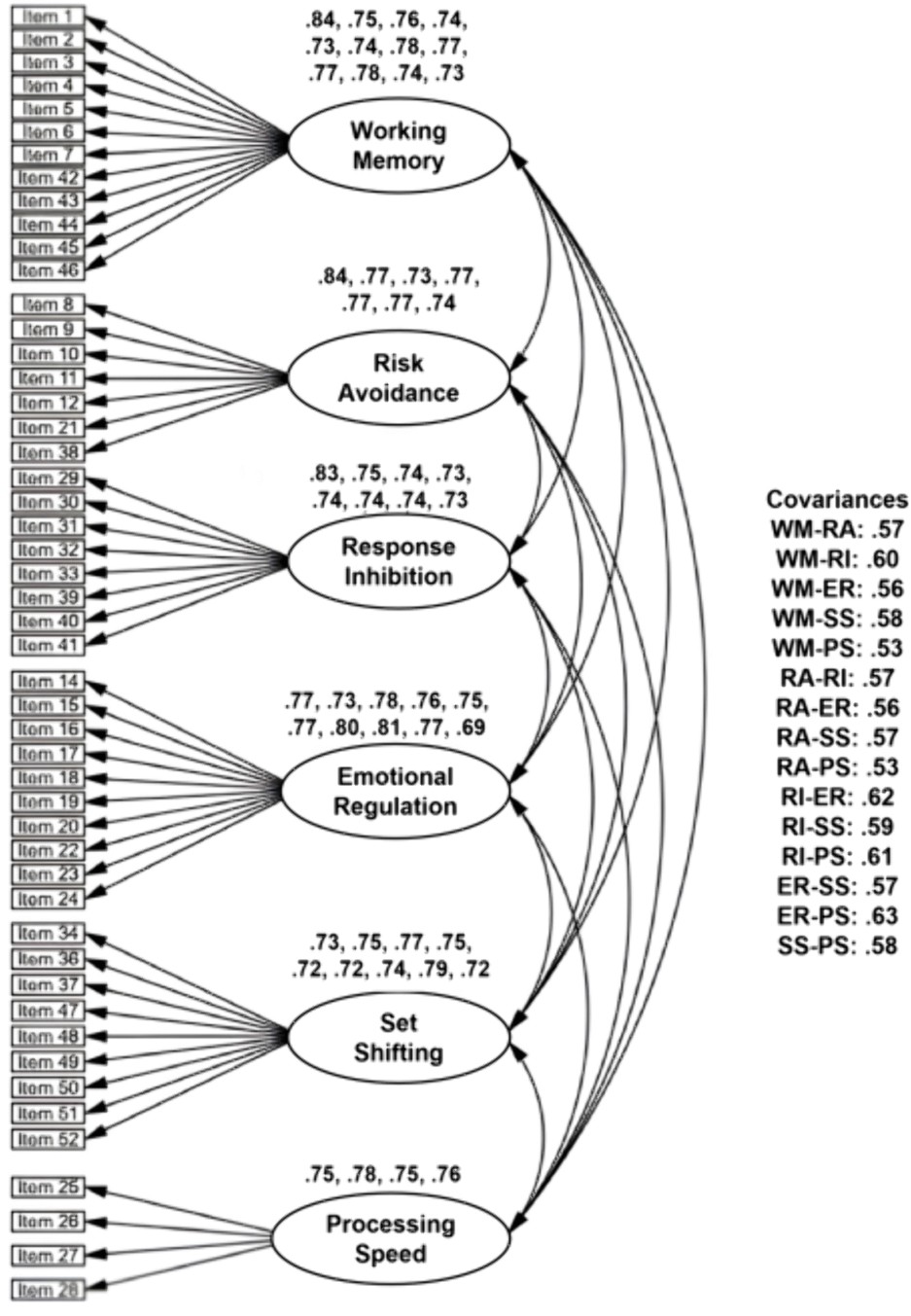

Note. All values are Standardised estimates outputs. The points are the factor

loading values for each path line.

**Fig 1. First order verification outcome.** All values are standardised estimates outputs. The points are the factor loading values for each path line.

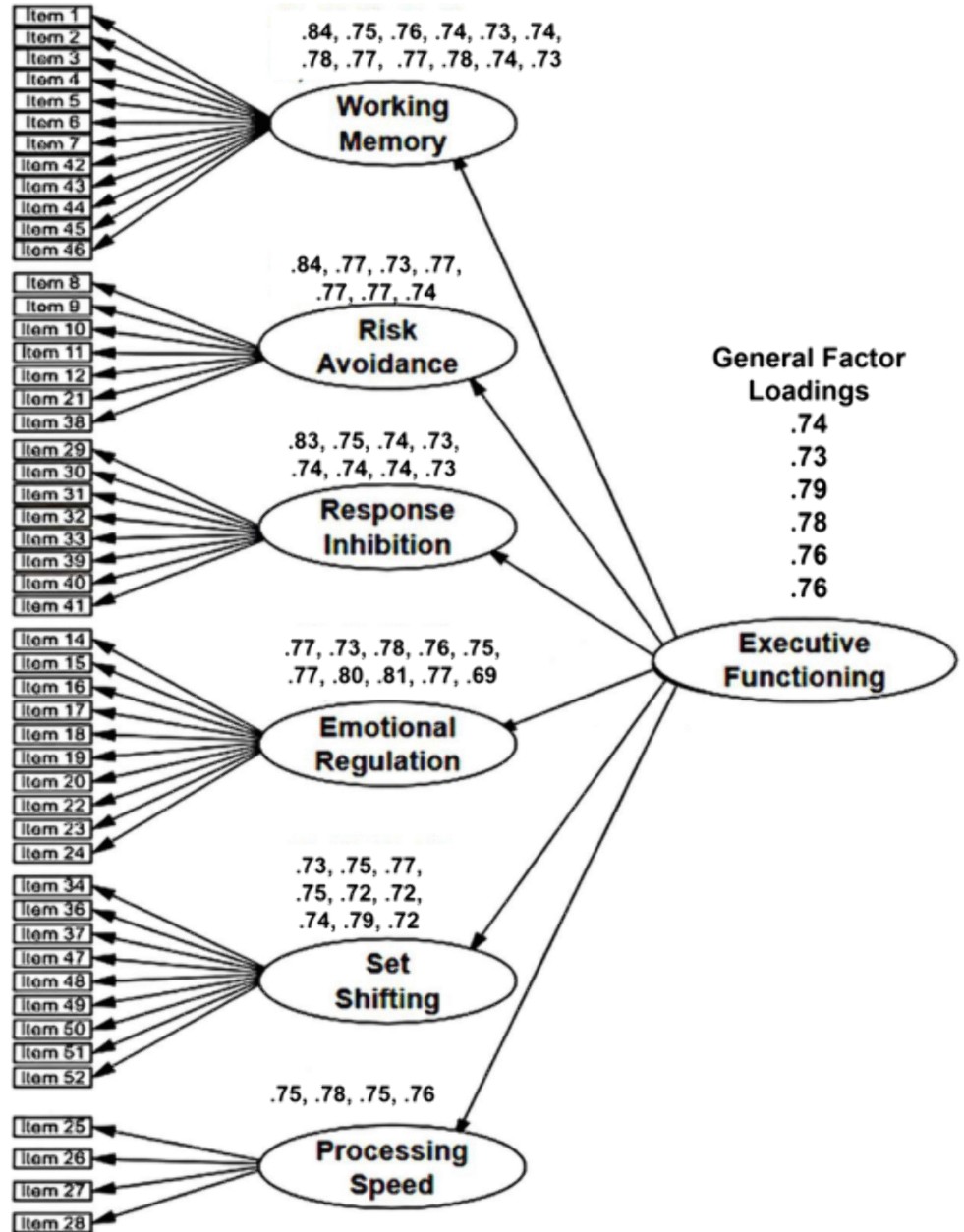

*Note*. All values are Standardised estimates outputs. The points are the factor loading values for each path line.

**Fig 2. Second order verification outcome.** All values are standardised estimates outputs. The points are the factor loading values for each path line.

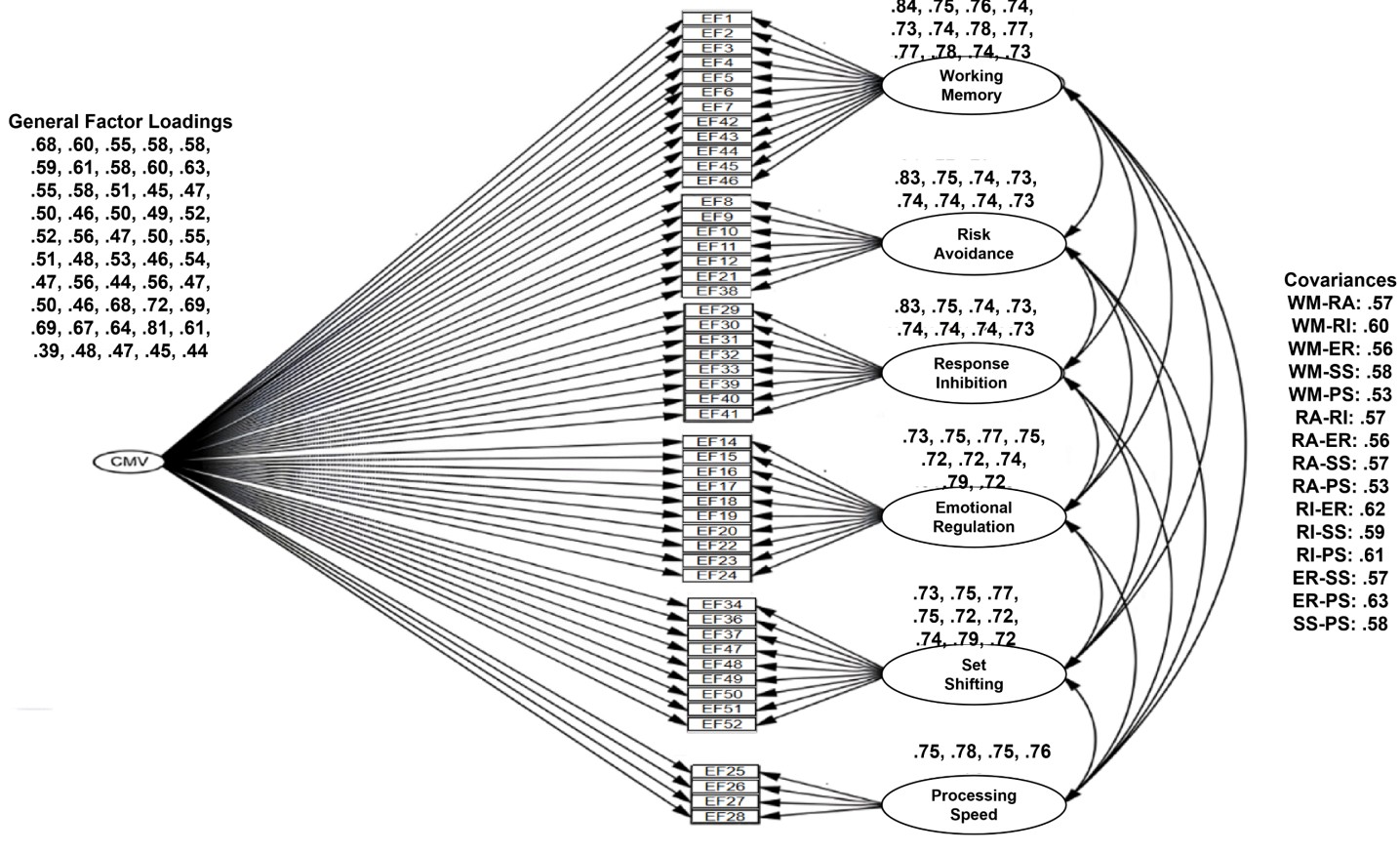

**Fig 3. Common method variance outcome.**

**Table 4. Fit Index Comparison on First&Second Order of Executive functioning Measured Model.**

| Model Comparison | CMIN/df | GFI | AGFI | RMSEA | TLI | CFI |
|---|---|---|---|---|---|---|
| FOV | 1.431 | .908 | .899 | .026 | .973 | .975 |
| SOV | 1.430 | .907 | .899 | .026 | .973 | .974 |
| FOV+CMV | 1.357 | .916 | .904 | .024 | .978 | .980 |

*Note.* FOV – first order model; SOV – second order model; CMV – common method variance; CMIN – Chi-square; CMIN/df – Chi-square/ df; GFI – Goodness of Fit Index; AGFI – Adjusted Goodness of Fit Index; RMSEA – Root Mean Square Error of Approximation; CFI – Comparative Fit Index; TLI – Tucker-Lewis Coefficient.

population. This outcome provides strong support for conceptualizing executive functioning as a higher-order construct, underpinned by six distinct yet cohesive sub-dimensions.

## Discussion

This study successfully adapted and validated the Executive Functioning Scale (EFS) for the Malaysian university student population, culminating in a psychometrically sound 50-item instrument. The central finding was the robust empirical support for a hierarchical, six-factor model of executive functioning. Through sequential exploratory and confirmatory factor

analyses, we demonstrated that while EF comprises six distinct yet correlated dimensions—including Working Memory, Set Shifting, and Response Inhibition—these components are parsimoniously explained by a single, higher-order latent construct. The resulting Malaysian-adapted EFS exhibited strong psychometric properties, including high internal consistency and excellent discriminative validity, confirming its utility as a reliable and valid instrument for assessing executive functioning within this unique cultural and educational landscape.

The significance of these findings is twofold. Theoretically, the confirmation of a second-order hierarchical model within a non-Western population provides compelling cross-cultural support for the prevailing "unity and diversity" model of executive functioning. This framework posits that while specific EF components are dissociable, they draw upon a common, underlying cognitive resource [2]. Our results empirically substantiate this model, suggesting that the macro-level cognitive architecture of EF possesses a degree of universality. Practically, this study directly addresses the critical psychometric void and the challenge of cultural bias raised earlier. Our findings robustly demonstrate that, following a rigorous cultural adaptation process, the EFS can be validly and reliably applied to the Malaysian university student population. This achievement moves beyond a mere technical validation; it provides local educators, clinicians, and researchers with an indispensable, evidence-based tool, resolving the untenable choice between unvalidated Western instruments and informal translations [13]. More fundamentally, this study serves as a compelling case study for the scientific adaptation of psychological assessment tools in non-Western cultural contexts. By systematically ensuring linguistic, conceptual, and contextual equivalence, our methodology offers a replicable blueprint for mitigating the "false deficits" that can arise from cultural incongruence, thereby promoting more equitable assessment practices. In this way, our work contributes to the global pursuit of educational fairness and inclusivity by providing a practical pathway to ensure cognitive assessments support, rather than penalize, students from diverse cultural backgrounds.

Our findings are broadly consistent with the extensive body of literature that conceptualizes executive functioning as a multidimensional construct. The emergence of distinct factors such as Working Memory [17], Set Shifting [18], and Response Inhibition [19] aligns with dominant theoretical models of EF, reinforcing their cross-cultural applicability. Furthermore, the superior fit of the hierarchical model resonates with prior structural equation modeling studies in Western contexts that have similarly identified a general EF factor [20]. Our study, however, also introduces important nuance. Whereas many studies converge on three-factor models, our analysis yielded a more differentiated six-factor solution [21]. This divergence may be an artifact of the EFS instrument's breadth, which is designed to capture a wider array of EF-related behaviors. Alternatively, it could reflect subtle cultural variations in how cognitive control behaviors are partitioned [22]. For instance, the identification of two distinct "Risk Avoidance" factors might suggest that risk assessment is not a unitary concept in this population but is instead context-dependent—a hypothesis that warrants deeper investigation.

The robust support for a hierarchical structure invites speculation on the underlying cognitive and neural mechanisms. It is plausible that the higher-order EF factor represents the functional capacity of a domain-general, frontoparietal neural network, which provides the foundational resources for cognitive control [8]. The first-order factors, in turn, may reflect the differential recruitment and weighting of specific nodes within this broader network to meet the demands of particular task sets. For example, Response Inhibition tasks might more heavily tax the orbitofrontal cortex [23], while Working Memory would depend more on the dorsolateral prefrontal cortex [24]. In the cognitively demanding university environment, students must constantly allocate this general EF resource to manage competing academic priorities (Set Shifting), retain complex material (Working Memory), and suppress distractions (Inhibition) [25]. The observed statistical structure, therefore, likely mirrors the functional reality of how a unified cognitive control system is flexibly deployed to solve distinct, real-world challenges.

## Limitations

Despite the strength of our findings, several limitations warrant acknowledgment. First, our sample was confined to university students, a demographic that may possess higher-than-average cognitive abilities and is not representative of

the broader Malaysian young adult population. This necessarily constrains the generalizability of our findings. Second, this study relied exclusively on self-report data, which captures perceived executive functioning rather than objective performance and is thus susceptible to social desirability and metacognitive biases [26]. Third, the cross-sectional design precludes inferences regarding the developmental trajectory of EF or causal relationships between EF and academic outcomes [27]. Finally, while we established the scale's validity for the overall Malaysian student sample, we did not conduct measurement invariance testing across the nation's distinct ethnic subgroups (e.g., Malay, Chinese, Indian), a crucial step for ensuring its equitable deployment in this multicultural context.

A further limitation of the present study is its exclusive reliance on a quantitative methodology. While the adapted EFS demonstrated robust psychometric properties, this approach, by its nature, cannot fully capture the subjective, culturally-mediated interpretations of the constructs being measured. Future research would be substantially enriched by adopting a mixed-methods design. Qualitative approaches, such as semi-structured interviews or focus groups, would allow for a deeper exploration of how Malaysian students conceptualize and navigate EF challenges in their own terms. This would provide invaluable context to the psychometric data, illuminating the cultural nuances behind response patterns and yielding a more holistic understanding of executive functioning within this specific population.

## Future directions

These limitations illuminate clear pathways for future research. To broaden generalizability, subsequent validation studies should engage more heterogeneous samples, including non-student populations and individuals across a wider age spectrum in Malaysia. A critical next step is to establish the scale's convergent validity by correlating EFS scores with performance-based EF measures (e.g., the Stroop test, WCST), a multi-method approach that would yield a more holistic assessment of the construct. Longitudinal designs are essential for mapping the development of EF throughout students' academic careers and for establishing its predictive validity for long-term outcomes, such as GPA and employability. Most urgently, future work must address measurement invariance to confirm that the scale functions equivalently across Malaysia's major ethnic groups, thereby ensuring its fair and unbiased application in both research and practice.

## Conclusion

In conclusion, this study offers more than a simple psychometric validation; it provides a foundational instrument that enables a new wave of culturally-attuned research in a historically underrepresented region. By establishing a valid and reliable measure of executive functioning, this work paves the way for a more sophisticated understanding of the cognitive factors shaping academic and personal success among Malaysian university students. This scale is poised to become a critical asset for advancing evidence-based educational and psychological practice throughout Malaysia.

## Supporting information

**S1 File. Minimal Data Set.** The raw data for this study can be found in this file.
(SAV)

**S2 File. Scale Items.** The English version of the scale, having undergone localised validity and reliability testing, is detailed in this document.
(DOCX)

## Acknowledgments

The authors thank the participating students and the expert panel for their assistance with translation and adaptation. We also acknowledge the Faculty of Education at Universiti Kebangsaan Malaysia (UKM) for providing the necessary administrative support and ethical oversight to facilitate this investigation.

## Author contributions

**Conceptualization:** Muhammad Syawal Amran.

**Data curation:** Hao Yin.

**Investigation:** Hao Yin.

**Methodology:** Muhammad Syawal Amran.

**Software:** Hao Yin.

**Supervision:** Muhammad Syawal Amran.

**Validation:** Hao Yin.

**Writing – original draft:** Hao Yin.

**Writing – review & editing:** Hao Yin.

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
