## [Decision Letter · Decision Letter 0]

22 Dec 2025

Dear Dr. Yin,

Thank you for submitting your manuscript to PLOS ONE. After careful consideration, we feel that it has merit but does not fully meet PLOS ONE’s publication criteria as it currently stands. Therefore, we invite you to submit a revised version of the manuscript that addresses the points raised during the review process.

https://journals.plos.org/plosone/s/submission-guidelines#loc-laboratory-protocols . Additionally, PLOS ONE offers an option for publishing peer-reviewed Lab Protocol articles, which describe protocols hosted on protocols.io. Read more information on sharing protocols at . Additionally, PLOS ONE offers an option for publishing peer-reviewed Lab Protocol articles, which describe protocols hosted on protocols.io. Read more information on sharing protocols at https://plos.org/protocols?utm_medium=editorial-email&utm_source=authorletters&utm_campaign=protocols ..

We look forward to receiving your revised manuscript.

Kind regards,

Maria José Nogueira, Ph.D.

Academic Editor

PLOS One

**Journal Requirements:**

1. When submitting your revision, we need you to address these additional requirements. Please ensure that your manuscript meets PLOS ONE's style requirements, including those for file naming. The PLOS ONE style templates can be found at https://journals.plos.org/plosone/s/file?id=wjVg/PLOSOne_formatting_sample_main_body.pdf and https://journals.plos.org/plosone/s/file?id=ba62/PLOSOne_formatting_sample_title_authors_affiliations.pdf 2. Your ethics statement should only appear in the Methods section of your manuscript. If your ethics statement is written in any section besides the Methods, please delete it from any other section. 3. We note that this data set consists of interview transcripts. Can you please confirm that all participants gave consent for interview transcript to be published? If they DID provide consent for these transcripts to be published, please also confirm that the transcripts do not contain any potentially identifying information (or let us know if the participants consented to having their personal details published and made publicly available). We consider the following details to be identifying information:- Names, nicknames, and initials- Age more specific than round numbers- GPS coordinates, physical addresses, IP addresses, email addresses- Information in small sample sizes (e.g. 40 students from X class in X year at X university)- Specific dates (e.g. visit dates, interview dates)- ID numbers Or, if the participants DID NOT provide consent for these transcripts to be published:- Provide a de-identified version of the data or excerpts of interview responses- Provide information regarding how these transcripts can be accessed by researchers who meet the criteria for access to confidential data, including:a) the grounds for restrictionb) the name of the ethics committee, Institutional Review Board, or third-party organization that is imposing sharing restrictions on the datac) a non-author, institutional point of contact that is able to field data access queries, in the interest of maintaining long-term data accessibility.d) Any relevant data set names, URLs, DOIs, etc. that an independent researcher would need in order to request your minimal data set. For further information on sharing data that contains sensitive participant information, please see: https://journals.plos.org/plosone/s/data-availability#loc-human-research-participant-data-and-other-sensitive-data If there are ethical, legal, or third-party restrictions upon your dataset, you must provide all of the following details (https://journals.plos.org/plosone/s/data-availability#loc-acceptable-data-access-restrictions):a) A complete description of the datasetb) The nature of the restrictions upon the data (ethical, legal, or owned by a third party) and the reasoning behind themc) The full name of the body imposing the restrictions upon your dataset (ethics committee, institution, data access committee, etc)d) If the data are owned by a third party, confirmation of whether the authors received any special privileges in accessing the data that other researchers would not havee) Direct, non-author contact information (preferably email) for the body imposing the restrictions upon the data, to which data access requests can be sent 4. If the reviewer comments include a recommendation to cite specific previously published works, please review and evaluate these publications to determine whether they are relevant and should be cited. There is no requirement to cite these works unless the editor has indicated otherwise. 

**Additional Editor Comments:**

Authors must improve manuscript according to reviewers recommendations.

Reviewers' comments:

**Comments to the Author**

1. Is the manuscript technically sound, and do the data support the conclusions?

Reviewer #1: Yes

Reviewer #2: Yes

2. Has the statistical analysis been performed appropriately and rigorously?

Reviewer #1: Yes

Reviewer #2: Yes

3. Have the authors made all data underlying the findings in their manuscript fully available?

Reviewer #1: Yes

Reviewer #2: Yes

4. Is the manuscript presented in an intelligible fashion and written in standard English?

Reviewer #1: Yes

Reviewer #2: Yes

**Reviewer #1:**  Cultural Bias in Cognitive Assessment Cultural Bias in Cognitive Assessment

Most executive functioning tools, including EFS, were developed in Western contexts. They often assume cultural norms about communication, problem-solving, and self-regulation that may not align with Asian cultural values (e.g., collectivism, hierarchical respect, indirect communication). This can lead to misinterpretation of abilities or false deficits. This issue can be better developed in the study.

Impact on Academic and Mental Health Support

Executive functioning is linked to academic success, stress management, and adaptability. If assessments are culturally biased, interventions may be ineffective or even harmful because they don’t reflect students’ real challenges.

Equity and Inclusivity in Higher Education

Universities increasingly aim for global standards. Using tools that ignore cultural nuances undermines fairness and can perpetuate stereotypes about cognitive performance across cultures.

Improving transcultural validity involves adaptation and validation, not just translation. Here are key strategies:

a) Linguistic and Conceptual Equivalence

Translate items using forward-backward translation.

Ensure concepts (e.g., “planning ahead”) have the same meaning in the target culture.

b) Cultural Adaptation of Scenarios

Modify examples to reflect local academic and social contexts.

Avoid idioms or culturally specific behaviors that don’t resonate globally.

c) Psychometric Validation

Conduct factor analysis to confirm that the scale measures the same constructs across cultures.

Test measurement invariance (configural, metric, scalar) to ensure comparability.

d) Include Local Norms and Expert Input

Engage local psychologists and educators in item review.

Collect normative data from diverse student populations.

e) Mixed-Methods Approach

Combine quantitative validation with qualitative interviews to capture cultural interpretations of executive functioning.

Please revue all this aspects.

**Reviewer #2:** Title: Psychometric Validation and Cultural Adaptation of the Executive Functioning Scale in Malaysian University StudentsTitle: Psychometric Validation and Cultural Adaptation of the Executive Functioning Scale in Malaysian University Students

Summary

I would like to congratulate the authors on addressing a timely and highly relevant topic. It was a pleasure to review this manuscript, which tackles an important gap in the literature by providing a culturally adapted and psychometrically validated measure of executive functioning for a non-Western context.

The manuscript presents the cultural adaptation and psychometric validation of the Executive Functioning Scale (EFS) for Malaysian university students. The study responds to a significant gap in the literature by offering a culturally appropriate instrument, supported by a rigorous methodological design and a large sample. Overall, the manuscript is clear, well structured, and reports robust psychometric results, aligning well with the standards of methodological rigor and transparency required by PLOS ONE.

Major Comments

No substantive methodological limitations were identified that would compromise the validity of the results or the conclusions. The study design, cultural adaptation procedures, psychometric analyses (EFA and CFA), and interpretation of the findings are appropriate and well justified. The comparison between competing factor models and the retention of the hierarchical model based on the principle of parsimony are particularly well supported.

Minor Comments

1. Cultural adaptation process

The authors are encouraged to explicitly clarify:

o the number of experts involved in the bilingual expert panel;

o whether consensus was reached through a formal procedure (e.g., Delphi method) or through consensus-based discussion.

This clarification would further strengthen methodological transparency.

o

2. Presentation of Table 3: Table 3 would benefit from improvements in presentation to enhance clarity and interpretability.

**Do you want your identity to be public for this peer review?** For information about this choice, including consent withdrawal, please see our For information about this choice, including consent withdrawal, please see our Privacy Policy .

Reviewer #1: **Yes:** Raul CordeiroRaul Cordeiro

Reviewer #2: No

---

## [Author Response · Author response to Decision Letter 1]

31 Dec 2025

Response to Editor

Regarding the requested citations for Table 1 and Figure 3 in the main text, we have highlighted these in red within the document titled "Revised Article with Changes Highlighted", on pages 7 and 15 respectively. These were correctly cited in the original version. Please refer to the red highlights on these two pages within that document. Thank you.

Response to Reviewers

Dear Editor and Reviewers,

Thank you for the opportunity to revise our manuscript, "Psychometric Validation and Cultural Adaptation of the Executive Functioning Scale in Malaysian University Students" (MS ID: [PONE-D-25-59203]). We are deeply grateful to you and the esteemed reviewers for providing such insightful and constructive feedback. The comments have been invaluable in helping us to strengthen the manuscript. We have addressed all points raised by the reviewers and detail these revisions below in a point-by-point response.

Response to Reviewer #1

We sincerely thank Reviewer #1 for their comprehensive and thoughtful review. We particularly appreciate his emphasis on the critical issue of cultural bias in cognitive assessment, which has been instrumental in enhancing the theoretical framing and practical implications of our study.

1. Comment on Developing the Issue of Cultural Bias, Its Impact, and Implications for Equity:

The reviewer correctly pointed out that the manuscript could more thoroughly develop the problem of cultural bias inherent in Western-developed tools, its downstream impact on academic and mental health support, and its broader implications for equity in higher education.

Response: We are grateful for this critical suggestion. To more robustly frame the rationale for our study, we have incorporated a new, comprehensive paragraph into the Introduction section (now the second paragraph). This section now explicitly discusses the WEIRD-centric origins of most EF tools, the concept of cultural dissonance, the risk of generating "false deficits," and the profound consequences of biased assessment for educational equity and inclusivity, thereby directly addressing the reviewer's primary concern.

2. Comment on Key Strategies for Improving Transcultural Validity (Points a, b, c, d):

The reviewer outlined a set of key best-practice strategies for cross-cultural adaptation, including a) ensuring linguistic and conceptual equivalence, b) adapting scenarios for cultural relevance, c) conducting rigorous psychometric validation, and d) incorporating local norms and expert input.

Response: We thank the reviewer for articulating these essential methodological standards. We are pleased to confirm that our study's methodology did, in fact, incorporate these steps. To make this explicit and to enhance methodological transparency, we have added a new, dedicated paragraph within the Methods section. This new section now systematically details our procedures, framing them using the reviewer's excellent recommendations:

Regarding points (a) & (b): We now explicitly describe our use of a standardized forward-backward translation protocol to establish linguistic equivalence and provide concrete examples (e.g., revising the "baking" item to a "research protocol" scenario) to illustrate how we ensured conceptual and contextual equivalence.

Regarding point (c): We reinforce that the purpose of the sequential EFA and CFA, detailed in the "Construct Validity" subsection, was precisely to achieve psychometric validation and confirm the scale's structural integrity within the Malaysian context.

Regarding point (d): We have clarified that an expert panel, comprising two local educational psychologists and one linguist, was engaged to review the adapted items to ensure alignment with local norms and cultural appropriateness.

3. Comment on a Mixed-Methods Approach (Point e):

The reviewer astutely suggested that a mixed-methods approach would be a valuable direction for future research.

Response: This is an excellent suggestion for deepening the field's understanding. As our current study's scope was focused on the quantitative psychometric validation of the EFS, we have incorporated this recommendation into our manuscript. Specifically, we have added a new paragraph to the Limitations section acknowledging the study's reliance on quantitative methods and explicitly proposing a mixed-methods design as a critical pathway for future research to explore the nuanced cultural interpretations of executive functioning.

Response to Reviewer #2

We are greatly encouraged by Reviewer #2’s highly positive assessment of our manuscript. We thank reviewer for his supportive comments on the study's rigor and for his helpful and constructive suggestions for minor revisions.

1. Comment on the Cultural Adaptation Process:

The reviewer requested clarification on the number of experts involved in the review panel and the process by which consensus was achieved.

Response: We thank the reviewer for this important point on methodological transparency. We have revised the new "Cultural Adaptation of the Executive Functioning Scale" subsection within the Methodology. The text now clearly states that the expert panel comprised "two local educational psychologists and one linguist." We have also added that consensus regarding item wording and cultural appropriateness was reached through iterative, consensus-based discussion among the panel members and the research team.

2. Comment on the Presentation of Table 3:

The reviewer noted that the presentation of Table 3 could be improved for clarity and interpretability.

Response: We are grateful for this feedback on data presentation. To enhance the clarity and interpretability of Table 3, we have reformatted it. The revised table now presents only the item numbers corresponding to each factor, rather than the full item text. This significantly improves readability and aligns with standard reporting practices for factor analysis results.

We are confident that these revisions have substantially improved the manuscript and thoughtfully address all the concerns raised. We look forward to your and the reviewers' positive evaluation of the revised submission.

Sincerely,

All authors.

---

## [Decision Letter · Decision Letter 1]

7 Jan 2026

Psychometric Validation and Cultural Adaptation of Executive Functioning Scale in Malaysian University Students

PONE-D-25-59203R1

Dear Dr. Hao Yin

We’re pleased to inform you that your manuscript has been judged scientifically suitable for publication and will be formally accepted for publication once it meets all outstanding technical requirements.

Kind regards,

Maria José Nogueira, Ph.D.

Academic Editor

PLOS One

Additional Editor Comments (optional):

The authors adequately address the reviewers' concerns.

The manuscript is suitable for acceptance.

Reviewers' comments:

Reviewer's Responses to Questions

**Comments to the Author**

Reviewer #1: All comments have been addressed

Reviewer #2: All comments have been addressed

2. Is the manuscript technically sound, and do the data support the conclusions?

Reviewer #1: Yes

Reviewer #2: Yes

3. Has the statistical analysis been performed appropriately and rigorously?

Reviewer #1: Yes

Reviewer #2: Yes

4. Have the authors made all data underlying the findings in their manuscript fully available?

Reviewer #1: Yes

Reviewer #2: Yes

5. Is the manuscript presented in an intelligible fashion and written in standard English?

Reviewer #1: Yes

Reviewer #2: Yes

Reviewer #1: No special comments. The observations have been well adressed and the paper is suitable to be acepted.

Reviewer #2: (No Response)

**Do you want your identity to be public for this peer review?** For information about this choice, including consent withdrawal, please see our For information about this choice, including consent withdrawal, please see our Privacy Policy .

Reviewer #1: **Yes:** Raul CordeiroRaul Cordeiro

Reviewer #2: No

---

## [Editor Report · Acceptance letter]

PONE-D-25-59203R1

PLOS One

Dear Dr. Amran,

I'm pleased to inform you that your manuscript has been deemed suitable for publication in PLOS One. Congratulations! Your manuscript is now being handed over to our production team.

Kind regards,

on behalf of

Professor Maria José Nogueira

Academic Editor

PLOS One